# Building an Indigenous-Led Evidence Base for Smoking Cessation Care among Aboriginal and Torres Strait Islander Women during Pregnancy and Beyond: Research Protocol for the Which Way? Project

**DOI:** 10.3390/ijerph18031342

**Published:** 2021-02-02

**Authors:** Michelle Bovill, Catherine Chamberlain, Jessica Bennett, Hayley Longbottom, Shanell Bacon, Belinda Field, Paul Hussein, Robert Berwick, Gillian Gould, Peter O’Mara

**Affiliations:** 1School of Health and Medicine, University of Newcastle, UON, Callaghan, NSW 2308, Australia; jessica.bennett@newcastle.edu.au (J.B.); gillian.gould@newcastle.edu.au (G.G.); peter.omara@newcastle.edu.au (P.O.); 2Hunter Medical Research Institute, Newcastle, NSW 2305, Australia; 3Judith Lumley Centre, La Trobe University, Bundoora, VIC 3086, Australia; c.chamberlain@latrobe.edu.au; 4Waminda- South Coast Women’s Health and Welfare Aboriginal Coorporation, Nowra, NSW 2541, Australia; HayleyLongbottom@waminda.org.au; 5Nunyara Aboriginal Health Unit, Gosford, NSW 2250, Australia; shanell.bacon@health.nsw.gov.au; 6Yerin-Eleanor Duncan Aboriginal Health Centre, Wyong, NSW 2259, Australia; belinda.field@yerin.org.au (B.F.); paul@yerin.org.au (P.H.); 7Tamworth Aboriginal Medical Centre, Tamworth, NSW 2340, Australia; RobertB@tams.org.au

**Keywords:** community based research, community engagement, smoking cessation, co-design, smoking during pregnancy

## Abstract

Strong and healthy futures for Aboriginal and Torres Strait Islander people requires engagement in meaningful decision making which is supported by evidence-based approaches. While a significant number of research publications state the research is co-designed, few describe the research process in relation to Indigenous ethical values. Improving the health and wellbeing of Aboriginal and Torres Strait Islander mothers and babies is crucial to the continuation of the oldest living culture in the world. Developing meaningful supports to empower Aboriginal and Torres Strait Islander mothers to quit smoking during pregnancy is paramount to addressing a range of health and wellbeing outcomes. Aboriginal and Torres Strait Islander women have called for non-pharmacological approaches to smoking cessation during pregnancy. We describe a culturally responsive research protocol that has been co-designed and is co-owned with urban and regional Aboriginal communities in New South Wales. The project has been developed in line with the AH&MRC’s (Aboriginal Health & Medical Research Council) updated guidelines for ethical research with Aboriginal and Torres Strait Islander communities. Ethics approvals have been granted by AH&MRC #14541662 University of Newcastle HREC H-2020-0092 and the Local Health District ethics committee 2020/ETH02095. Results will be disseminated through peer reviewed articles, community reports, infographics, and online social media content.

## 1. Introduction

In my language (MB), Wiradjuri, we have a word, walan, meaning strong, and walan-ma-rra, meaning to make strong. Strength, to be walan (strong), is not just about physical strength, but also encompasses emotional and spiritual strength which are connected to our country, kin, culture and community. The walan (strong) health and wellbeing of Aboriginal and Torres Strait Islander women and their babies is crucial to the continuation of the oldest continuing culture in the world. In Australia, there is much to be done to walan-ma-rra (to make strong) the health and wellbeing of Aboriginal and Torres Strait Islander people, whose current health is a direct result of colonisation, dispossession, genocide and ongoing racism that is experienced throughout generations and remains today. Walan-ma-rra (to make strong) the health of Aboriginal and Torres Strait Islander mothers and babies, a wide range of important research is currently being conducted including birthing [1,2,3], and culturally safe care [4,5,6,7,8].

Our research is in the area of maternal health and smoking cessation. Specifically, we are developing empowering cessation supports for Aboriginal and Torres Strait Islander women based on the needs and desires of Aboriginal and Torres Strait Islander women. This research builds on previous research [9,10,11] and ongoing dialogue with Aboriginal and Torres Strait Islander women and communities. Our project is called “Which Way?” This paper reports a culturally responsive research protocol that has been developed in partnership and co-ownership with urban and regional Aboriginal communities in New South Wales. The project is outlined in reference to the CONSIDER statement [12] to offer transparency of ethical health research practice in line with Indigenous recognised values and principles. The CONSIDER statement, developed in 2019, is an international checklist for the reporting of health research involving Indigenous people, developed to strengthen research praxis and advance Indigenous health outcomes [12]. The Which Way? project is a reflection of reciprocal relationships with Aboriginal and Torres Strait Islander women and community. This is the project women asked for [11], led by a Wiradjuri woman in partnership with Aboriginal communities who are leading the research process and co-own the data collected. There remains a current gap in clinical trials that are built on Aboriginal community knowledge, expertise and wisdom. A bridge needs to be built between Indigenous knowledge science and those of Western scientific knowledge to enhance health outcomes [13] with recognition of sovereignty of Aboriginal and Torres Strait Islander people and its correlation to rights to have power over our health. Walan-ma-rra (to make strong) the health and wellbeing of Aboriginal and Torres Strait Islander peoples has been noted in Australia throughout several campaigns, plans and strategies that highlight health disparities such as the Closing the Gap strategy launched in 2008 [14].

This strategy has failed Aboriginal and Torres Strait Islander people due to lack of self-determination and adequate funding [15]. In 2020, the Prime Minister noted that the failure over the past 12 years to reduce the gap in life expectancy between Indigenous and non-Indigenous Australians was attributed to a lack of true partnership with Aboriginal and Torres Strait Islander people [16]. In 2020, peak bodies in Aboriginal and Torres Strait Islander health have refreshed the Closing the Gap strategy in partnership with the Australian Federal Government, all States and Territories, and Australian Local Government Association, with 16 new targets. In both the previous Closing the Gap targets, and the new reform, addressing intergenerational health has a focus on maternal health. There are ongoing targets to address the high prevalence rates of smoking during pregnancy among Aboriginal and Torres Strait Islander women [17]. In addition, a National Tackling Indigenous Smoking program has also prioritised smoking during pregnancy in recent program reviews and acknowledges the current gap in services and support specifically targeted to Aboriginal and Torres Strait Islander women during pregnancy [18].

While data is often presented from a deficit position, stating Aboriginal and Torres Strait Islander pregnant mothers are three times more likely to smoke during the first 20 weeks of pregnancy than non-Aboriginal women (45% vs. 13% age standardized) [19], it is reported less often that these Aboriginal and Torres Strait Islander mothers do make frequent quit attempts during pregnancy [10]. High rates of quit attempts have been reported in the Aboriginal and Torres Strait Islander community with the National Tobacco Strategy 2012–2018 midpoint report, compared to the general population [19]. The report stated that Aboriginal and Torres Strait Islander people are more likely than non-Aboriginal people to make a quit attempt but less likely to succeed [19]. There is an indication here that support services may not be meeting the needs of this population. There has been a 9.8% absolute decrease in Aboriginal and Torres Strait Islander smokers over the past 15 years [20]. Developing appropriate programs to support cessation should build on these successes and privilege Aboriginal and Torres Strait Islander ways of knowing (epistemology), being (ontology) and doing (axiology) [21]. 

## 2. Study Aims

The Which Way? project aims to co-develop an Indigenous-led evidence base for a smoking cessation intervention to support Aboriginal and Torres Strait Islander mothers to be smoke-free during pregnancy and beyond. Which Way? Smoking Cessation project is a mixed-methods, community-led research project that partners with four Aboriginal communities in New South Wales, Australia, over three years to explore what Aboriginal and Torres Strait Islander women’s needs and desires are for holistic smoking cessation during pregnancy. The project asks, “Which way?” which is a commonly used phrase among Aboriginal and Torres Strait Islander people meaning what, where, how. By asking “Which way?”, the project is asking Aboriginal and Torres Strait Islander women to develop and direct the research on what they want and need to support smoking cessation during pregnancy and beyond with a focus that is non-pharmacological.

Primary objectives:To determine Aboriginal and Torres Strait Islander women’s preferences for non-pharmacological approaches to smoking cessation.Explore non-pharmacological approaches being utilised in Aboriginal communities and/or explore how desired approaches identified in study 1 would be implemented appropriately for Aboriginal and Torres Strait Islander women.Explore Health Provider attitudes and beliefs towards non-pharmacological approaches to smoking cessation.

Once the project has met the above objectives the project team and research governance committee will co-develop an intervention and conduct a pilot for feasibility. 

## 3. Methods and Analysis

It must be noted that this research project was impacted by COVID-19. The research was originally designed to conduct qualitative studies first to inform quantitative work, however, due to community safety, the research was re-ordered by the research governance committee. This protocol reflects this reorganised approach with an acknowledgement that this was not the originally intended process.

### 3.1. Research Prioritisation

The Which Way? project is an extension of lead investigators’ PhD studies exploring “culturally responsive approaches to smoking cessation among Aboriginal and Torres Strait Islander pregnant women”. Through community consultation and qualitative data collection over three years, Aboriginal and Torres Strait Islander women continued to express a need for non-pharmacological approaches, “natural” approaches to smoking cessation that built on community strengths, interests and worldviews [9,11,22]. The Which Way? project engaged local community through the shared priority and interest in addressing new and innovative approaches to smoking cessation for Aboriginal and Torres Strait Islander women during pregnancy and beyond. The project was presented at conferences and community meetings between June and November 2019 by the lead investigator to articulate the purpose of the study and generate interest in the study by Aboriginal Health Services. An information flyer was developed and shared on social media and through peak organisations and networks to engage Aboriginal Health Services in New South Wales to partner in the study. Each interested community contacted the project team and after consultation(s) with appropriate staff and information shared with community boards, approvals were granted, and partnerships established. The project lead, in time, took over six months and was conducted without funding. This process included the development of an Aboriginal Research Governance Committee, drafting of an agreed term of reference and developing the project plan, and drafting ethics applications.

### 3.2. Research Governance

All health research conducted with Aboriginal and Torres Strait Islander peoples in New South Wales must adhere to the Aboriginal Health & Medical Research Council’s (AH&MRC) ethical guidelines. AH&MRC key principles include: net benefits to Aboriginal people and communities, Aboriginal Community Control of Research, cultural sensitivity, reimbursement of costs, and enhancing Aboriginal skills and knowledge. An updated, (version 2) AH&MRC Ethical Guidelines: Key Principles was released this year (2020) [23], which also now ensures Aboriginal governance of the research are embedded in the research with three possible categories: direct Aboriginal Community Controlled Health Service (ACCHS) involvement, non-ACCHS involvement and Aboriginal researchers. AH&MRC ethical guidelines state that Aboriginal people have a right to make decisions about research, as well as have their involvement, support and consent sought. Aboriginal Governance must be incorporated into all stages of the project [23]. AH&MRC advise that if there is no direct ACCHS involvement in the research, that an Aboriginal Reference Group is to be established. In acknowledgement of the importance of Aboriginal oversight guided by local Aboriginal knowledges and wisdom, we developed an Aboriginal Research Governance Committee with partnering communities to strengthen our research and transparency processes.

Each partnering Aboriginal Health Service is a full partner and co-owner of the research. Each service has supplied formal letters of community support and hold a seat on the Aboriginal Research Governance Committee (ARGC). The ARGC is established to oversee the research process including research design, implementation, analysis and reporting. The ARGC was established as each partnering community service offered a letter of support and confirmed their long-term commitment to the project. The ARGC, with the research team, developed a Terms of Reference for the group, which is a drafted, shared document that can change over time, as the project changes, staff changes, or competing demands and priorities change. The Terms of Reference offers a tool for transparency of the reciprocal relationship between the research team and the partnering communities. The Terms of Reference outlines the function of the committee as well as both the community’s requests of the Research Team and the Research Team’s requests from community. The research governance committee meets monthly via Zoom and are provided with brief written reports on project progress for staff to share with their service staff and community. The ARGC is chaired by the lead investigator (MB), an Aboriginal researcher.

The ARGC consult, design and direct all of the research conduct. The ARGC act as a community representative, ensuring that research processes, plans and reports are shared with relevant community and offer feedback and direction to the research team. The ARGC support directing and coordinating further community consultations, reporting back to community and methods of research dissemination that is appropriate and meaningful to each community. Ongoing, informal dialogue occurs throughout the project with members of the ARGC on community needs or interest in other research areas, updates on other projects or developments within the community or upcoming events. This is an organic and reciprocal relationship process that goes beyond the research project or even research itself. Rather, it is a representation of Indigenous epistemology (knowing), ontology (being), and axiology (doing).

### 3.3. Relationships

The Which Way? project partners with: Yerin Aboriginal Health Centre, Tamworth Aboriginal Medical Service, Waminda Aboriginal Women’s Health Service and Nunyara Aboriginal Health Unit for the Central Coast Local Health District. Each service invites interested staff members to engage as researchers on the project. In addition to the strong community partnership in the project described in the Governance section, the project engages all interested members of staff at the partnering site. The project is led by an Aboriginal woman with over 15 years’ experience working in partnership with Aboriginal and Torres Strait Islander communities in social work and health research. The lead investigator is offered guidance in the development and conduct of Aboriginal health research by two senior Aboriginal academics/clinicians (C.C. and P.O’M.). The project team engages Aboriginal and Torres Strait Islander leadership in health with 4 of the 5 investigators being Aboriginal. The project has been awarded funding from the NHMRC Early Career Researcher Award which was awarded based on the PhD findings and preliminary consultations, and the project was further refined through consultation and engagement. The funded proposal articulated a community-led approach with acknowledgement that this may vary the proposed research program. Further funding was awarded by the National Heart Foundation Aboriginal and Torres Strait Islander Award, which funds the studies described in this paper. With the support of the National Heart Foundation Aboriginal and Torres Strait Islander Award, an Aboriginal medical student has been employed as a Research Assistant on the project who is offered ongoing mentoring, training and opportunities to engage in the research.

The project has been developed in line with the AH&MRC updated guidelines for ethical research with Aboriginal and Torres Strait Islander communities [23]. Ethics approvals have been granted by AH&MRC (#14541662) University of Newcastle HREC (H-2020-0092) and the Local Health District ethics committee (2020/ETH02095).

### 3.4. Research Methodologies

Supportive and encouraging smoking cessation interventions for Aboriginal communities should build on Aboriginal values of resilience, empowerment and trust [24,25]. Building on the knowledge, expertise and wisdom of local Aboriginal community, the Which Way? project aims to inform the development and implementation of an intervention that is culturally responsive, offering new approaches to service delivery for Aboriginal women [26,27]. This research applies the Indigenist Research framework developed by Rigney to the health space to recognise and respect Indigenous ways of knowing, being and doing [28]. As such, this research acknowledges the need for self-determination and is undertaken with a link to political struggle with reference to the Uluru Statement from the Heart [29] and privileges the voices of Aboriginal and Torres Strait Islander people. By applying an Indigenist research framework, the study challenges the assumptions that inform the research, the framing of questions, and the approaches used, centreing the cultural knowledge and practices of the lead investigator as a Wiradjuri woman in the conduct of research.

The Which Way? project applies an Indigenist conceptual framework, builds on a Community Based Participatory Research paradigm [30], with acknowledgement of the Aboriginal Health and Medical Research Council’s (AH&MRC) updated ethical guidelines [23] to conduct Community-Based Indigenous Research, excluding the word ‘participatory’ and reflecting Indigenous ownership and leadership on the project. 

Figure 1 outlines the research framework. The research acknowledges the importance of building on ‘Gulbanha’ (knowledge), to ensure the research is relevant and meaningful to improving the health and wellbeing of Aboriginal and Torres Strait Islander people. Indigenous knowledge is both the process and the outcome of the research. At the core of the framework is the Research Governance, made up of Aboriginal Health Services staff, with embedded Indigenous research methodologies running through the work conducted, leading to the outcome of new knowledge. The brown lines represent the voices and experiences of Aboriginal and Torres Strait Islander people engaged in the research process.

### 3.5. Research Participation

The research project consists of three studies designed to address current gaps in the literature, that will inform the development of a pilot smoking cessation program.


Study 1: Online survey


The aim of the survey is to determine Aboriginal and Torres Strait Islander women’s preference for non-pharmacological approaches to smoking cessation. Drawing on our previous research and a comprehensive literature search, an online cross-sectional survey will be conducted to understand Aboriginal and Torres Strait Islander women’s quitting experiences and desires for a range of smoking cessation supports. Twenty cessation support options including face to face and online modes of delivery, support groups, exercise programs, phone app counselling, cultural programs and nicotine replacement therapy will be able to be selected. The survey will include Aboriginal and Torres Strait Islander women of reproductive age, 16 years and over, smokers and ex-smokers (any level of consumption) and exclude non-Aboriginal or Torres Strait Islander women and non-smokers. The survey will consider factors influencing choices of non-pharmacological options based on: age, use of Aboriginal Health Services, remoteness and heaviness of smoking.

Online surveys hosted on the secure REDCap survey software will be conducted using social media accounts that are created for the project, which will detail all necessary information on the study and the survey instrument. A Which Way? Facebook and Instagram page will be developed to share information on the project and project team as well as host regular information sharing in the space of Aboriginal and Torres Strait Islander health and wellbeing, along with the project. The survey will recruit through social media paid advertisements and snow-ball sampling through social networks.

The study team, including partnering Aboriginal Health Services, will promote the survey on their Facebook and Instagram pages and send throughout their networks. Partnering services will have an iPad available at the service for community members to complete the survey when attending the service. Aboriginal and Torres Strait Islander community Facebook and Instagram pages will be contacted by the research team and provided with an overview of the project, project team and communities, and asked to share information on their page if aligned with their values, without an incentive or payment. The survey will be conducted over a 3-month period and participants will be eligible to go in the draw for an iPad or reimbursements through the community service. A minimum of 385 women will be recruited.


Study 2: Yarning circles


The aim of the yarning circles is to explore what non-pharmacological approaches are being utilised in communities and how desired approaches identified in study 1 could be implemented appropriately for Aboriginal and Torres Strait Islander women in a culturally safe, engaging and empowering way. This part of the study will engage with Aboriginal and Torres Strait Islander women across the partnering Aboriginal communities in culturally appropriate qualitative research. Yarning methods will be used [31] to conduct yarning circles led by Aboriginal women (Aboriginal post-doctoral researcher (M.B.) and a staff member of the Aboriginal Health Service). Yarning is a method for qualitative research that has been used internationally with Indigenous groups [31]. Yarning is a credible and rigorous approach to establishing relationships with Indigenous participants in the research practice. Bessarab and Ng’andu 2010 describe the yarning method:

*Yarning in a semi-structured interview is an informal and relaxed discussion through which both the researcher and participant journey together visiting places and topics of interest relevant to the research study. Yarning is a process that requires the researcher to develop and build a relationship that is accountable to Indigenous people participating in the research*.[31]

Posters and social media posts will be promoted by the individual partnering Aboriginal Health Services with a local contact and project contact listed to recruit Aboriginal and Torres Strait Islander women of reproductive age, 16 years and over, smokers and ex-smokers (any level of consumption). Service staff will also discuss the research project with pregnant and non-pregnant smokers and ex-smokers when they attend clinic or community programs in an informal way, following the approach of research topic yarning [31] methodology to build community interest. Yarning circles will be held in partnership with each partnering Aboriginal Health Service. The coordination and promotion of the circles will be guided by each individual community.

Community members who have an interest in being involved in the yarning circle will attend the circle date and time that suits them, and the dates and locations are directed by the partnering communities. They will be welcomed in and offered tea, coffee, or a cold drink and reminded that no research commences until everyone understands the process and consents formally.

Yarning circles will follow a structured process of:

*Social yarning*—the researchers introduce themselves so that social and cultural positioning and relationality may be identified [32].

*Research topic yarning*—the researchers will explain the research topic, purpose, and actions to follow. Researchers will read out loud the content of the participant information sheet and provide each woman with an information sheet. Written informed consent is gathered during this yarning process as well as a general discussion on the researchers’ plan for the research, reporting and dissemination.

*Collaborative yarning*—For those participants that consent, the researchers (M.B. with a member of the ARGC for the site) will lead a recorded yarning circle with no structured questions but rather, domains of enquiring which include: experiences of quitting smoking, cessation supports that are not medications, and how might the community run the suggested program to engage and empower women to be smoke-free. The content shared during the social and research topic yarning will ignite conversations on domains of exploration before recorded collaborative yarning commences.

*Sharing a meal*—At the closing of the yarning circles, a lunch will be offered to the women. This process acts as an acknowledgement of women giving up their time to participate in the research as well as providing a culturally safe space for women and researchers to be able to reflect on the yarning circle conversations. The conversations will offer reflectivity for both the participants and researchers, being offered the time to converse, reflect and create notes or images [33].

Yarning circles will be audio recorded through the Notability app on a University- owned, password-protected iPad pro. The app can simultaneously record the audio session and allow images to be drawn on a large blank screen. Acknowledging art as a cultural practice, and that some women may prefer artistic expression as a process for conversation, this app allows the research practice to collect multiple forms of data at once. The iPad page through the app is endless so women can move and draw on the screen continuously over the yarning session time. The drawings will also be used for reflective practice with the ARGC and during data analysis. The Notability app allows the data to be immediately uploaded to the University’s secure OneDrive and can separate the audio and visual data. Audio will be transcribed by a professional transcription agency.

Yarning circles will go for approximately 2 h, but the exact length of time will depend on the communities’ needs and timelines. Yarning circles will be conducted with all partnering sites; due to the wide community reach of partnering sites, it is planned that each site will have 1–2 yarning circles with 10–12 women in each yarning circle. A total of 6–8 yarning circles with approximately N = 96 Aboriginal and Torres Strait Islander women will be conducted. Venues for the yarning circles will be the partnering Aboriginal Health Services or local community centres requested by the individual communities to address any possible transportation barriers. The circles will include Aboriginal and Torres Strait Islander women of reproductive age, 16 years and over, smokers and ex-smokers (any level of consumption) and will exclude non-Aboriginal or Torres Strait Islander women and non-smokers.


Study 3: Email survey


The aim of this cross-sectional survey is to explore Health Provider attitudes to non-pharmacological approaches and supports requested by Aboriginal women in study 1 to smoking cessation. This study will implement an emailed survey of Health Providers to understand their confidence in offering cessation support to Aboriginal and Torres Strait Islander women, as well as their attitudes and beliefs of non-pharmacological approaches to smoking cessation. The survey will be the first to report baseline knowledge in this area, as such, report descriptive statistics.

The survey will recruit registered Health Providers working with Aboriginal and Torres Strait Islander people (including but not limited to Aboriginal health workers and practitioners, general practitioners, nurses, midwives). Email survey recruitment will be conducted through promotion to peak organisations in Aboriginal and Torres Strait Islander Health Services via their listservs and newsletters. Survey data will be collected using secure REDCap software. The survey will include only registered Health Providers working with Aboriginal and Torres Strait Islander communities. A minimum of 385 Health Providers will be recruited.

### 3.6. Research capacity

The research project builds the skills and capacity of the lead investigator’s early career research in her first years of post-doctoral studies, mentored and supervised by senior Aboriginal academics/practitioners. Through the funding from the National Heart Foundation, further capacity-building is offered to an Aboriginal medical student working as a Research Assistant on the project. Through respectful relationships with partnering communities, two-way learning and knowledge sharing is offered on diverse community health needs, research design and implementation, and knowledge translation and health promotion. This two-way learning is reflected in this publication which includes authorship by members of the ARGC representing their ownership of the research project. 

### 3.7. Data Analysis

The Which Way? project acknowledges the rights of Aboriginal peoples to govern the creation, collection, ownership and application of their data, and has developed a data management plan through consultations with the partnering communities who will continue to monitor the implementation of the plan through the ARGC. We acknowledge the partnering communities as the knowledge holders and, as such, the owners of all data collected by and on their community. Ensuring the analysis process incorporates Indigenous world views and localised knowledges, a reflective process is embedded in each study and overseen by the ARGC. A data analysis plan is first drafted in consultation with ARGC and project investigators, and ongoing reflective analysis is offered through ARGC meetings which revisits analysis design and outcomes to allow localised strengths-based knowledge to be articulated. Preliminary findings will be presented to the ARGC for feedback and comment. Once the ARGC approves the analysis, final analysis and reporting will be conduct with the ARGC’s oversight and direct involvement. 

### 3.8. Research Dissemination

Through strong, long-term and reciprocal relationships, ongoing dissemination of findings can occur to all partners and communities. Acknowledging Aboriginal and Torres Strait Islander people to be active users of social media [34], this method of knowledge sharing will be used. Development of monthly updates is an important mechanism for accountability of the research team to communities, reporting on both the big and small changes, advances and achievements of the study. Monthly reports are tabled at internal ACCHS’ committees including board meetings, research community meetings and staff meetings. At the closing of each study, a community lunch will be held at each site where the research team will verbally present the results of the study and next steps for the research to the community. An ongoing knowledge translation plan will be reviewed by the research team and ARGC to ensure Indigenous governing bodies are informed on the project. The development of peer reviewed publications that arise from the study will engage the ARGC in co-authorship, ensuring communities are represented appropriately and acknowledging the wisdom, leadership and expertise of partnering communities in developing an Indigenous-led evidence basis for smoking cessation care.

### 3.9. Study Timeline

The project commenced in June 2019. COVID-19 impacted some delays in the project. 

Study 1 commenced July 2020 and closed in October 2020, study 2 yarning circles are anticipated to commence January through to February 2021 and study 3 will commence March through to June 2021. June to December 2021, the research team, in partnership with the research governance committee, will co-develop a pilot trial of a non-pharmacological smoking cessation program using the evidence gathered in the three studies. 

## 4. Discussion

This is a novel mixed-methods study that moves beyond the benevolence approach to research, and privileges Aboriginal and Torres Strait Islander people as knowledge holders. Through long-term, reciprocal partnerships with Aboriginal Health Services, the Which Way? project demonstrates how research practice can uphold Aboriginal and Torres Strait Islander ethics, and co-develop meaningful policy and practice change. This project builds on the limited knowledge of what Aboriginal and Torres Strait Islander women want to support and empower them to be smoke-free during pregnancy and beyond. Building on the needs and desires of women, it is envisioned that adherence to support strategies and resulting cessation outcomes can be achieved.

This study rose from the requests of Aboriginal and Torres Strait Islander women [24] and addresses an area of research not yet approached. Which Way? aims to walan-ma-rra (to make strong) Aboriginal and Torres Strait Islander women and their communities through developing a meaningful support program to empower smoke-free lives. The research process engages in Indigenist research, as Tuhiwai Smith states, that Indigenous control over research means that questions are “framed differently: priorities are ranked differently: problems are defined differently: and Indigenous people participate on different terms” [35]. This will be achieved by Aboriginal researchers partnering with Aboriginal communities, a research process acknowledging Aboriginal communities as knowledge holders and experts, Aboriginal community governance guiding the project, the development of localised evidence, community-level dissemination and knowledge exchange and the enactment of AH&MRC ethical guidelines. It is through Indigenist research that, we contend, real improvements to health outcomes can be achieved.

## Figures and Tables

**Figure 1 ijerph-18-01342-f001:**
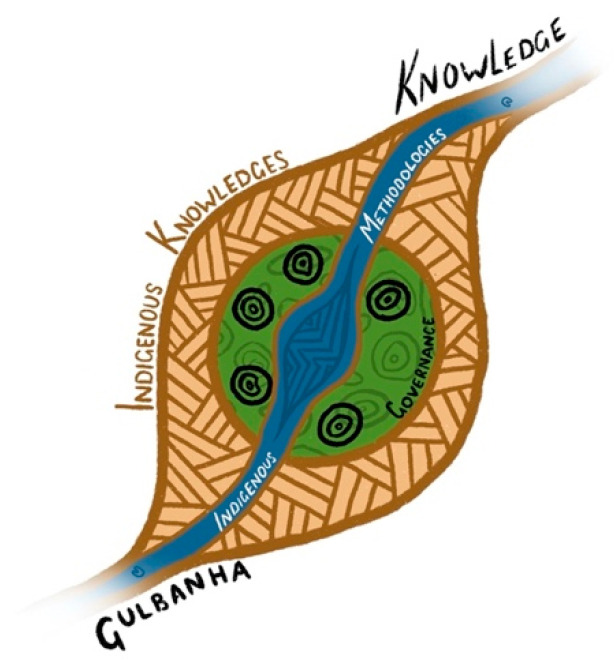
Outlines this research framework.

## Data Availability

No data collected as part of this research project is available for sharing. This is in line with Aboriginal and Torres Strait Islander ethical guidelines.

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
