# Peer review of "Building an Indigenous-Led Evidence Base for Smoking Cessation Care among Aboriginal and Torres Strait Islander Women during Pregnancy and Beyond: Research Protocol for the Which Way? Project"

_ijerph, 2021, doi:10.3390/ijerph18031342_

Round 1

Reviewer 1 Report

Please refer to attached mark up

n offering my review of the “Building an Indigenous-led evidence base for smoking cessation care among Aboriginal and Torres Strait Islander women during pregnancy and beyond: Research Protocol for the Which Way? Project” manuscript, I take an Indigenous lens that considers cultural worldviews underpinned by my area of expertise regarding Indigenous philosophies, methodologies and methods in the design and delivery of research. Importantly, an Indigenist paradigm and standpoint is central when peer reviewing this article.

The ‘Which Way’ project is a smoking and cessation initiative that has been co-designed and co-owned by Aboriginal and Torres Strait Islander women drawn from four communities of the state of New South Wales, Australia. The initiative is building on an earlier study that has previously secured support and engagement with these four communities. Funding similarly has been sourced to support the next iteration of the cessation project.

I note that ethic approval has been secured from the Aboriginal Health and Medical Research (AH&MRC), University of Newcastle, and Local Health District ethic committees. I further note that due to COVID-19, the research study has been re-designed to adjust for the unexpected events of a global pandemic.

The manuscript describes a culturally responsive and ethical process to the design and delivery of research with the knowledges of ways of knowing, being and doing held by Aboriginal and Torres Strait Islander women and the research team central to its approach.

Overall, this article is a worthy paper that makes significant contribution to gaps in scholarly knowledge. It contributes towards the evidence base regarding ethical processes of co-developing a project like the “Which Way” initiative with Aboriginal and Torres Strait Islander women including the governance and accountability mechanisms. Therefore, it is recommended that the paper should be accepted for publication subject to consideration of the proposed minor revisions indicated in the attached tracked mark ups.

Article contribution – Originality/Novelty including significance

Emphasising the need to have Aboriginal and Torres Strait Islander people’s ways of knowing, being and doing central to responses to support cessation of smoking amongst pregnant women represents the significance of this article because presently western sciences of ways of knowing, being and doing dominant public health responses to smoking cessation.  Critically, Indigenous led evidence base initiative is crucial if outcomes are to be fully achieved.

The paper provides an example of exemplary practice concerning engagement and negotiating with Aboriginal and Torres Strait Islander women that draws from their knowledges of ways to address the reduction of smoking during pregnancy. Additionally, a holistic response to smoking cessation during pregnancy goes beyond the typical cause and effect public health response. Specifically, co-developing and co-creating a non-pharmacological response is where evidence is lacking. The ‘Which way?” project will contribute towards such an evidence gap. Additionally, the ethical steps involved in how to co-develop an initiative such as the “Which way?” – the what, where and how – further adds to an evidence gap of working within a cultural context involving Indigenous people/communities. What strengthens this manuscript further in such approaches is the use of the

  1. CONSIDER statement
  2. AH&MRC ethical guidelines
  3. a governance committee made up of Aboriginal representation from the participating communities and health services who oversee and direct the research design, implementation, analysis, and reporting. Importantly, their knowledges and wisdom – an Indigenous epistemology, ontology, and axiology - is bought to the research process and they co-own the results, which is not often taken into consideration by non-Indigenous mainstream western sciences.
  4. The lead investigator is a Wiradjuri woman who is supported by two senior Aboriginal academics/clinicians that includes 4 of the 5 investigators being Aboriginal. The team is predominately Aboriginal which ensures that an indigenous worldview of ways of knowing, being and doing is complemented with those of participating communities.

Quality of presentation including English level

Overall, the manuscript is presented well with an appropriate English level. I saw enjoyed the introduction and the use of language to describe strong and to make strong including what is meant by which way.

Although there were parts in the paper which was ‘clunky’ that made the reader having to work hard to read the material, and this has been identified in the marked-up manuscript attached.

Scientific soundness

The use of Rigney’s Indigenist research framework including the CONSIDER statement and the demonstration of Indigenous ways of knowing, being and doing involving reciprocity, relationality, responsibility, reverence, and respect weaved throughout the paper makes the paper scientifically and ethically sound.

Interest to readers

This paper will be of interest to a wide audience in both western and Indigenous sciences including qualitative and quantitative researchers involving Indigenous peoples. This is because Indigenous research methodologies are complementing and sitting alongside of western science methodological approaches that makes a further contribution to the scholarly knowledge base.

Overall merit

This is a worthy paper for the reasons identified above.

Recommendations for further consideration

Accept paper for publication subject to consideration of the proposed minor revisions.

Author Response

A PDF of the proposed minor revisions for this reviewer is not visible on our end. We look forward to the opportunity to address these comments once the document has been sent through.

Reviewer 2 Report

Thank you for writing this important research protocol for your work on the Which Way project.  I thoroughly enjoyed reading your protocol, and found it to be very well written, clear concise and inclusive of all Indigenist methods/methodologies and principles that evidence the co-authors’ deep immersion and expertise in this topic area. After several reads and reflecting on the content, I cannot provide much feedback that could improve this work and recommend it for publication as is, after the correction of only several very minor typos/edits, which I have highlighted on the uploaded PDF of your manuscript. From my knowledge and understanding of Indigenist methodologies, (as described by Rigney and others) your paper showcases a highly respectful and comprehensive approach to this important work in reducing smoking rates for Aboriginal and Torres Strait Islander women during pregnancy.  Each stage of this work resonates so well with Indigenist methods.  Several dot points below provide a little feedback from my perspective. I have also made several positive comments on some highlighted text in the PDF, along with highlighting only a couple of minor typos/edits requiring correction:

  • Your on-line survey is an important inclusion here and the recruitment via snowballing through social networks honours the relational and connected ethos of Indigenous ways of knowing, being and doing.
  • Reading that 4 of the 5 investigators are Indigenous is both practical and theoretical validation of your study and showcases it’s rigour and value.
  • The only comment I make as regards the clarity of your paper is that for international audience, it may be wise to briefly define Yarning, for readers and researchers outside Australia. I note that you have cited Dawn Bessarab’s excellent paper which defines it, however I think that just a sentence or two informing the readers about Yarning as a legitimate and increasingly used data collection method, would help to further inform the research world about it.  I have not however made this as conditional on publication, just a suggestion, and I leave this up to your judgement and expertise.   
  • I particularly like the way the first author has used her voice in the introduction and included some Wirajuri words that walan-marra your paper and protocol. It is pleasing to see the (slow howwever ongoing) progress made over the past 2 decades on crossing  previous barriers to including Indigenous way of knowing being and doing into academic journals.  A paper I submitted was rejected and slammed a decade ago by several Australian medical journals for using the first-person voice in a paper with several co-authors.  We then cited relational accountability and the co-creation of new knowledge, as well as Indigenist principles (all of which strengthened our paper), and eventually got the paper published in a British medical journal.
  • Wishing you well in this important work and I look forward to reading future reports and results from your research

Author Response

Thank you for your considerate review.

Point 1: I cannot provide much feedback that could improve this work and recommend it for publication as is, after the correction of only several very minor typos/edits, which I have highlighted on the uploaded PDF of your manuscript.

Response 1: The minor edits have been corrected as per the uploaded document. This is a correction on line 66-67 and 322.

Point 2: The only comment I make as regards the clarity of your paper is that for international audience, it may be wise to briefly define Yarning, for readers and researchers outside Australia.

Response 2: Thank you for this suggestion. Additional information on yarning have been included from line 310-319.

Reviewer 3 Report

This is a useful protocol focused on the favorite way required by aboriginal women to achieve smoking cessation and to determine the preferred pharmacological approach.

Inclusion and exclusion criteria should be reported in the methods section. Please provide a statement about the type of study , if it is observational or retrospective.

A statistical method should be added explaining what the sample size will be and the statistical tests that will be performed. The aim described in research methodologies on line 240, page 5 should be moved to the study aims section. Please include in the project the tests that will be used to assess smoking cessation (urine cotinine or exhaled CO, etc)

In the methods  please clarify which non pharmacological approaches will be considered and which variables will be explored as factor influencing smoking habit(e.g comorbidities, mood, social status):

If psychological support, electronic cigarettes , nicotine replacement therapy.

Three references should be added to broaden the discussion :

-BMJ Open . 2019 Jun 4;9(6):e025293.

-Int J Environ Res Public Health. 2020 Nov 14;17(22):8432.

-Subst Abus. 2018;39(3):289-306. doi: 10.1080/08897077.2018.1439802

Author Response

Thank you for your consideration review. I have addressed your comments to the best of my ability whilst maintaining the overall position of the paper as an Indigenist methods protocol.

Point 1: Inclusion and exclusion criteria should be reported in the methods section. Please provide a statement about the type of study, if it is observational or retrospective.

Response 1: Inclusion and exclusion criteria have now been added on lines 280-283 & 385-392 & 405-406. This study is not the intervention so is neither observational or retrospective, no additional explanation has been added about the type of study beyond the statements in line 238-270.

Point 2: A statistical method should be added explaining what the sample size will be and the statistical tests that will be performed. The aim described in research methodologies on line 240, page 5 should be moved to the study aims section. Please include in the project the tests that will be used to assess smoking cessation (urine cotinine or exhaled CO, etc).

Response 2: Sample sizes have been added to each study and descriptive areas of inquiry line 305-306 & 389-390 & 413-414. This is not an intervention so no smoking cessation validation is required beyond self report. Additional statistical methods have also been added to lines 404-405 however statistical methods are not described in detail as these are subject to ongoing community consultation and direction as described in the “data analysis” section L 432-447.

Point 3: In the methods please clarify which non pharmacological approaches will be considered and which variables will be explored as factor influencing smoking habit(e.g comorbidities, mood, social status): If psychological support, electronic cigarettes, nicotine replacement therapy.

Response 3: Cessation supports are now described on lines 280-287. Factors influencing choice have now been added to line 286-288.

Point 4: Three references should be added to broaden the discussion:

-BMJ Open . 2019 Jun 4;9(6):e025293.

-Int J Environ Res Public Health. 2020 Nov 14;17(22):8432.

-Subst Abus. 2018;39(3):289-306. doi: 10.1080/08897077.2018.1439802

Response 4: Thank you for your suggestion. The discussion does not currently discuss smoking cessation but rather the overall position of the paper which articulates Indigenist research practice. It has not been edited at this stage as it will extend the paper and focus.